# Sexual Dysfunction in Women and Men with Psoriasis: A Cross-Sectional Questionnaire-Based Study

**DOI:** 10.3390/medicina58101443

**Published:** 2022-10-13

**Authors:** Kamila Kędra, Kinga Janeczko, Izabela Michalik, Adam Reich

**Affiliations:** 1Students’ Scientific Circle of Experimental Dermatology, Department of Dermatology, Institute of Medical Sciences, Medical College of Rzeszow University, 35-055 Rzeszow, Poland; 2Department of Dermatology, Institute of Medical Sciences, Medical College of Rzeszow University, 35-055 Rzeszow, Poland

**Keywords:** quality of life, psoriasis, sexual problems, stigmatization

## Abstract

*Background and objectives:* Psoriasis can lead to feelings of stigmatization, hinder social functioning, and impair quality of life. Psoriasis can also affect sexual activity, but there is still little research on this topic. The present study investigated whether and to what extent psoriasis, its severity, location and extent of skin lesions affect sexual dysfunction. *Materials and Methods:* A total of 45 women and 64 men aged 18 to 73 years hospitalized for psoriasis exacerbations were included in the study. Psoriasis severity, as assessed by the Psoriasis Area and Severity Index (PASI), ranged from 0.2 to 65 points (mean: 17.0 ± 14.9 points). After collecting demographic and clinical data, each subject was asked to complete the Dermatology Life Quality Index, the 11-item Sexual Life Questionnaire and the International Index of Erectile Function (only men). *Results:* Our study found that more than 90% of the patients surveyed felt at least a slight unattractiveness due to psoriasis. In approximately 80% of the subjects, the skin lesions at least occasionally affected their sex life, and more than 50% at least sometimes avoided sexual contact. The location of psoriasis, particularly in the genital area (*p* = 0.01), on the face (*p* = 0.03) and hands (*p* = 0.05), also had a significant impact on the level of sexual problems. Psoriasis has a significant impact on the quality of life (QoL), and a deterioration in QoL was strongly correlated with sexual dysfunction (r = 0.6, *p* < 0.001), PASI scores (r = 0.36, *p* < 0.001), self-assessment of psoriasis severity and location of psoriatic lesions. *Conclusions:* Psoriasis leads to various limitations, especially in the sphere of sexual life. Patients with psoriasis feel stigmatized, have lowered self-esteem and consequently experience significant sexual problems. Awareness of the co-occurring psychological aspect of psoriasis and the routine use of validated scales in dermatology practice should contribute to the rapid identification of patients with sexual dysfunction.

## 1. Introduction

Psoriasis is a chronic immune-mediated disease with a genetic background, triggered by psychological and/or environmental factors, with the highest incidence in high-income Western Europe countries [1]. It can pose a significant burden and take a toll on many areas of a patient’s life. Psoriasis cannot be treated as a disease confined to the skin, but often coexists with or may be associated with various comorbidities, such as metabolic syndrome, diabetes, atherosclerosis, respiratory disease, cognitive problems, heart disease or arterial hypertension [2,3,4,5]. Approximately 20–30% of patients with psoriasis may also have psoriatic arthritis, which often causes significant musculoskeletal pain that significantly impairs quality of life (QoL) [6,7]. In addition, people with psoriasis frequently experience skin pain and itch, which contributes to reduced QoL [8]. Patients with the disease also experience disfiguring skin lesions and symptoms such as desquamation. Thus, the disease can have a significant impact on the patient’s social functioning and cause feelings of stigma [9]. 

Stigmatization is defined as a trait that disrupts interpersonal relations and is a cause of discrimination and social alienation [9]. This phenomenon is common in patients with psoriasis—even more so than in patients with other visible skin diseases, largely due to the lack of public knowledge about the disease. One study found that only approximately 10% of psoriasis patients experience no stigma, while the rest experience varying degrees of it [10]. Psoriasis negatively affects patients’ lives in many ways and can contribute to non-compliance with treatment [11,12]. Moreover, many studies have linked psoriasis to depression, emotional stress and reduced quality of life. Nearly 80% of people with psoriasis show symptoms of depression, and more than 70% had anxiety [13]. In addition, the characteristic lesions present on the skin lead to lower self-esteem and difficulties in social interactions. It is worth noting that appearance is one of the elements of self-confidence and plays an important role in interpersonal relationships. For this reason, psoriatic skin lesions can hinder sexual initiation and negatively affect sex life. One previous study found that more than 90% of men felt unattractive and embarrassed during psoriasis flare-ups [12]. Some studies have suggested that psoriasis also affects men’s sexual dysfunction. Psoriasis patients have been found to have more problems with sexual dysfunction than the healthy population [14,15]. 

To date, several studies have been conducted on the sexuality of men with psoriasis, but there are still few studies on these problems in women, for example, sexual dysfunction was observed in 48.7% of patients studied in one study of 102 British women with psoriasis [2]. Since sexuality is an important aspect of life, and there are still few studies on the subject, we believe that more research should be conducted. In our study, we investigated whether and to what extent psoriasis, its severity, location and extent of skin lesions affect the severity of sexual dysfunction in both sexes. Although similar studies have been conducted in the past [16,17], awareness of sexual problems in psoriasis patients is still limited and often ignored by treating physicians. We believe that providing further data on this important aspect of psoriasis patients’ well-being, with a particular focus on the role of feeling of stigma, would enrich knowledge about this aspect of psoriasis. 

## 2. Materials and Methods

### 2.1. Patients and Methods

Forty-five women and sixty-four men (1:1.4) with psoriasis aged 18 to 73 years (mean 48.0 ± 13.4 years) participated in a cross-sectional, questionnaire-based study of sexual dysfunction. Participants were consecutively recruited from among women and men appearing for a routine visit or hospitalized for an exacerbation of psoriasis at the Department of Dermatology in Rzeszów (Poland) between January 2020 and December 2021. Inclusion criteria for the study were: consenting adult; having active psoriatic lesions on the skin; and speaking the Polish language. All patients underwent a thorough dermatological examination, which included an assessment of the clinical severity of psoriasis based on the Psoriasis Area and Severity Index (PASI). Among all participants, 91 had plaque-type psoriasis (83.5%), 14 (12.8%) had palmoplantar pustular psoriasis; 16 had concomitant nail psoriasis (14.7%), and 28 had scalp psoriasis (25.7%). Active psoriatic arthritis was present in 18 (16.5%) subjects (Table 1). 

### 2.2. Assessment of Sexual Problems, Erectile Dysfunction and Quality of Life

After a thorough medical history-taking, all patients were asked to complete the Polish version of the Dermatology Life Quality Index [18] and the 11-item Sexual Life Questionnaire, used by our group in a previous study [12]. Patients were required to choose one of the following responses: “never”, “occasionally”, “sometimes”, “often” and “all the time” (scored from 0 to 4) for questions relating to psychosocial and emotional problems (7 questions—Table 2) or “not at all”, “yes, a little”, “yes, markedly”, “yes, very much” for questions regarding levels of shame and attractiveness (4 questions scored 0 to 3—Table 3). In addition, each of the participating male was asked to complete a 5-item version of the International Index of Erectile Dysfunction (IIEF-5). All patients consented to participate in the study.

### 2.3. Statistical Analysis

All results were statistically analyzed using Statistica 13.0 software (Statsoft, Kraków, Poland). Continuous variables were described as mean (M), standard deviation (SD), maximum and minimum values. Normal distribution was assessed using the Kolmogorov–Smirnov test. Comparisons between groups were performed using Student’s t-test, analysis of variance and multiple regression analysis. Pearson’s correlation test was used to calculate correlations between variables. *p*-value less than 0.05 were considered statistically significant.

## 3. Results

### 3.1. Psoriasis Severity, Location of Psoriatic Lesions and Comorbidities

At the time of study, patients had psoriasis from 1 to 55 years (mean disease duration: 19.0 ± 12.1 years). According to PASI, the clinical severity of psoriasis ranged from 0.2 to 65 points (mean: 17.0 ± 14.9 points). Eighty (74.8%) patients reported up to two psoriasis flares per year, seven (6.5%) reported more than two flares per year, and in twenty (18.7%) patients, skin lesions persisted throughout the year. Two patients did not answer this question. Regarding visible and sensitive skin areas, psoriatic lesions were present on the face in 39 (35.8%) patients, on the hands in 62 (56.9%), on the scalp in 80 (73.4%), on the nails in 55 (50.5%) and in the genital area in 50 (45.9%) participants. A positive family history of psoriasis was found among 45 (41.3%) of the included subjects. A total of 48 (44.0%) psoriasis patients had other comorbidities: cardiovascular disease was documented in 26 (23.9%), diabetes in 16 (14.7%), prostate adenoma in 2 (1.8%), malignancy in 3 (2.8%) and other diseases (endocrine, neurological, autoimmune diseases) in 23 (21.1%) patients (Table 1). 

### 3.2. Sexual Problems in Patients with Psoriasis

More than 90% of the patients surveyed felt at least slightly unattractive due to psoriasis. A significant proportion of respondents (>60%) said they felt at least markedly embarrassed when skin lesions were present on visible areas of the body. More than 70% of respondents admitted that they felt embarrassed when skin lesions were located in the genital area. Nearly 40% of the participants declared that they felt ashamed often or even all the time when they were with their sexual partner. Among 80% of the respondents, skin complaints at least occasionally affected their sex life. In addition, many participants (>50%) responded that they avoided sexual contacts at least sometimes because of psoriasis. A significant number of participants also declared that their sexual activity had decreased at least somewhat because of the skin problem (Table 2 and Table 3).

Regarding clinical parameters, there was a small but statistically significant correlation between disease severity as assessed by the PASI and sexual problems (r = 0.2, *p* = 0.04, Table 4). Even more significant was the subjective perception of disease severity. Again, subjects who consider their psoriasis severe had significantly more sexual problems (mean scoring: 22.1 ± 9.9 points) than subjects who considered their psoriasis moderate (mean scoring: 16.2 ± 9.1 points) or mild (11.5 ± 9.0 points, *p* = 0.001, Table 4). In addition, the level of sexual problems was significantly influenced by the location of psoriasis, particularly in the genital area (*p* = 0.01), on the face (*p* = 0.03) and on the hands (*p* = 0.05) (Table 4). Considering demographics, those who were divorced or separated showed the highest level of sexual problems (*p* < 0.05; Table 4). The other parameters studied had no significant effect on the observed level of sexual problems in psoriasis (Table 4).

### 3.3. Erectile Problems

Erectile dysfunction was screened using the IIEF-5 test. Based on the results, erectile problems (scores < 20 points out of 25) were suspected in 21 (34.4%) subjects. Erectile problems were naturally associated with increasing age, but, interestingly, were even more associated with low education (*p* < 0.001) and low income (*p* < 0.01) (Table 4). Erectile dysfunction was also more likely associated with comorbid cardiovascular disease. Importantly, the severity of psoriasis, location or duration of psoriasis, did not appear to have a significant effect on erectile function in men with psoriasis (Table 4). 

### 3.4. Correlation between Quality of Life and Sexual Problems

Among the 108 (99.1%) patients who completed the DLQI, 49 (45.4%) showed an extremely large effect (DLQI scoring: 21–30 points), and another 28 (25.9%) showed a very large effect (DLQI > 20 points) of psoriasis on their QoL. Only 10 (9.3%) subjects showed no impact of psoriasis on their QoL, 11 showed a small effect (DLQI: 2–5 points), and the remaining 10 (9.3%) patients showed a moderate impact on QoL (DLQI: 6–10 points). QoL was highly correlated with sexual dysfunction (r = 0.6, *p* < 0.001) (Figure 1), but not with erectile problems as assessed by the IIEF-5 (r = 0.01, *p* = 0.99). In addition, QoL was significantly correlated with PASI scores (r = 0.36, *p* < 0.001), self-assessment of psoriasis severity, income level (*p* < 0.05) and location of psoriatic lesions (Table 4). The other parameters studied had no effect on the measured QoL level (Table 4). Based on multiple regression analysis, sexual problems (β = 0.46, *p* < 0.001), self-assessment of psoriasis severity (β = 0.21, *p* < 0.01), PASI score (β = 0.19, *p* = 0.02) and the presence of genital psoriatic lesions (β = 0.18, *p* = 0.03) were found to be independent parameters affecting patients’ QoL. 

## 4. Discussion

According to previous studies, psoriasis has been portrayed not as a disease that is limited only to the skin, but as a complex health disability that has a major impact on many spheres of patients’ lives [10,11,13]. Psoriasis is believed to affect patients’ psychological well-being and lead to reduced self-esteem. Recent studies have shown that psoriasis can also influence the spheres of interpersonal relations and impede sexual initiation [2,8,12]. It has been suggested that psoriasis may have a greater impact on sexual dysfunction than neurodermatitis [19]. Importantly, feeling of stigma among patients with psoriasis appears to contribute to lowered self-esteem, alienation, and thus, feelings of shame and sexual avoidance [20]. One study found that, according to 51.1% of respondents, society is intolerant to patients with psoriasis, and 59.1% said they felt discriminated against because of their altered skin condition [21]. The location of psoriatic lesions was a clinically significant determinant of the level of stigma, and patients with visible lesions scored significantly higher on the Feelings of Stigmatization Questionnaire than those with invisible ones [10]. It has been suggested that emotional problems, depression and a higher suicide risk are associated with perceptions of higher psoriasis severity [22]. Interestingly, feeling stigmatized was the most influential predictor of depressive symptoms for psoriatic patients and accounted for 33% of the variance [20]. The presence of depression has also been identified as the major psychological factor linking psoriasis to sexual dysfunction [23]. In an earlier study, depression and sexual dysfunction were found to be more prevalent in people with psoriasis compared with controls, and the greater impact of psoriasis on sex life was associated with the lower sense of attractiveness and the lower quality of life [12]. Consistent with these findings, our observations documented a close relationship between the presence of psoriasis skin lesions and lowered self-esteem leading to avoidance of social contacts and feelings of shame in front of a sexual partner. 

It has been observed that psoriasis has a negative impact on quality of life and sexual health, especially when the lesions were located in the genital area, and affected women in particular [24]. According to our study, the feeling of being unattractive and the severity of the disease significantly affected and hindered sexual life among patients with psoriasis. In the current study, the importance of the location of psoriatic lesions was observed—our study showed a significant association between the presence of psoriasis on the face, genital area and hands, and the occurrence of sexual problems, but we did not observe significant differences regarding sexual dysfunction between men and women. Importantly, sexual dysfunction was the most important independent predictor of QoL impairment in the patient population studied. 

Regarding erectile dysfunction, we found that approximately one-third of patient with psoriasis may suffer from erectile problems, but this was independent of psoriasis severity. Bardazzi et al. even found a higher prevalence of erectile dysfunction among milder forms of psoriasis (56.7%) compared with patients with severe psoriasis (46.7%) [25]. A recent systematic review of the literature found that five of eight studies observed a higher risk of erectile dysfunction in patients with psoriasis compared with healthy men [23]. It was reported that erectile dysfunction was significantly increased in patients with more advanced psoriasis [26]. The presence of erectile dysfunction was associated with age, comorbidities such as cardiovascular disease, diabetes, arterial hypertension, neurological problems, and prostate disease. However, it has been suggested that the mechanism of erectile dysfunction in psoriasis appears to be not only organic, but may also be psychogenic, significantly affecting sex life [9]. Nevertheless, reducing the risk of major comorbidities in psoriasis should also prevent the occurrence of erectile dysfunction. It has been suggested that biologic treatment may be helpful in reducing erectile dysfunction and appears to be effective in improving sex life among patients with psoriasis. Biologic treatment is effective in eliminating psoriatic lesions, and thus, may be associated with increased self-esteem and reduced feeling of shame [23]. In contrast, some drugs used to treat psoriasis, such as etretinate/acitretin and methotrexate, have been associated with sexual and erectile dysfunction [27]. In addition, other medications used to treat comorbidities, such as antidepressants, muscle relaxants, antihypertensive drugs, H1 antagonists, anxiolytics and thiazides, can also negatively affect sexual function [28]. However, the occurrence of sexual dysfunction is also associated with other factors such as smoking, alcohol abuse, poor diet, obesity and low levels of physical activity [27]. Therefore, biological treatment should be combined with changes in lifestyle and daily habits. Of particular importance is that in the future, each psoriasis patient will be treated in a more personalized way, with targeted therapies not only focusing on skin and joint lesions, but also treating comorbidities and reducing the perceived burden of disease.

## 5. Conclusions

Patients with psoriasis feel stigmatized, have lowered self-esteem and consequently experienced significant sexual problems. Mental disorders such as depression, other comorbidities, lowered self-esteem, smoking and particular medications can negatively affect sexual well-being among psoriasis patients. A comprehensive approach focused not only on the treatment of skin lesions, but also on the patient’s overall mental state, plays a crucial role in proper psoriasis treatment. Often co-occurring with psoriasis, depression, anxiety disorders and lowered self-esteem, as well as the skin lesions themselves, lead to limitations in the sexual life sphere. The routine use of validated scales in dermatology practice should contribute to the rapid identification of patients with sexual dysfunction. Awareness of the comorbid psychological aspect of psoriasis can help clinicians provide holistic care, and thus, more effective therapy for this entity.

## Figures and Tables

**Figure 1 medicina-58-01443-f001:**
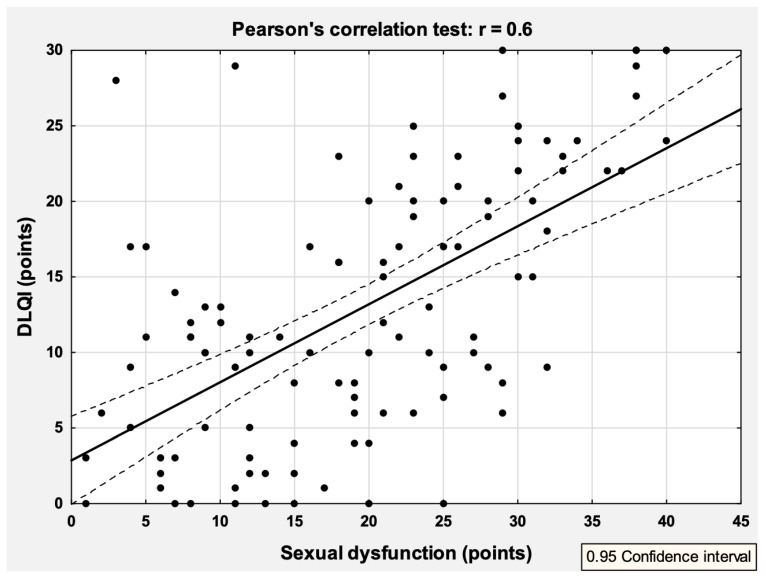
Correlation between sexual dysfunction and the quality of life (DLQI—Dermatology Life Quality of Index).

**Table 1 medicina-58-01443-t001:** Patients’ characteristics (Min—minimum, Max—maximum, PLN—Polish zloty).

Age (Years) [Mean ± Standard Deviation (Min–Max)]	48.0 ± 13.4 (18–76)
Psoriasis duration (years) [mean ± standard deviation (min-max)]	19.0 ± 12.1 (1–55)
Education	
Primary school	10 (9.2%)
Secondary school	30 (27.5%)
High school	38 (34.9%)
University	31 (28.4%)
Domicile	
Countryside	46 (42.2%)
A city with up to 50,000 inhabitants	26 (23.9%)
A city with 50–150,000 inhabitants	9 (8.3%)
A city with over 150,000 inhabitants	28 (25.6%)
Earnings	
Earnings lower than PLN 2000	36 (33.0%)
Earnings from PLN 2000 to 5000	61 (56.0%)
Earnings from PLN 5000 to 10,000	9 (8.2%)
Earnings above PLN 10,000	3 (2.8%)
Marital status	
Single	23 (21.1%)
Married	65 (59.6%)
Divorced/Separated	13 (11.9%)
Widow/Widower	6 (5.5%)
Comorbidities	
Cardiovascular disease	26 (23.9%)
Diabetes	16 (14.7%)
Prostate adenoma	2 (1.8%)
Malignancy	3 (2.8%)
Other (endocrine, neurological, autoimmune diseases)	23 (21.1%)
Psoriasis outbreaks during the year	
One or two per year	72 (66.1%)
More than two per year	8 (7.3%)
All the time (no remissions)	17 (15.6%)
No recurrences of the disease (first psoriasis episode)	6 (5.5%)
No answer	6 (5.5%)
Specific location of skin lesions	
Face	39 (35.8%)
Hands	62 (56.9%)
Genital area	50 (45.9%)
Scalp	80 (73.4%)
Nails	55 (50.5%)
Type of psoriasis *	
Plaque-type psoriasis	91 (83.5%)
Palmoplantar pustular psoriasis	14 (12.8%)
Psoriatic arthritis	18 (16.5%)
Nail psoriasis	16 (14.7%)
Scalp psoriasis	28 (25.7%)
Family history of psoriasis	
Yes	45 (41.3%)
No	64 (58.7%)
Patient’s assessment of the psoriasis severity	
Mild	8 (7.3%)
Moderate	32 (29.4%)
Severe	69 (63.3%)

* One patient might have more than one psoriasis subtype.

**Table 2 medicina-58-01443-t002:** Results on psychosocial and emotional problems in men and women with psoriasis related to their sexual life.

	Never	Occasionally	Sometimes	Often	All the Time	Missing Data
Has your skin condition ever affected your sex life?	19 (17.4%)	20 (18.3%)	21 (19.3%)	32 (29.4%)	14 (12.8%)	3 (2.8%)
Do you think other people considered your skin problem as contagious disease?	9 (8.3%)	11 (10.1%)	35 (32.1%)	33 (30.3%)	19 (17.4%)	2 (1.8%)
Do you avoid social contacts because of your skin problem?	24 (22.0%)	24 (22.0%)	22 (20.2%)	25 (22.9%)	12 (11.0%)	2 (1.8%)
Do you avoid sexual contacts because of your skin problem?	24 (22.0%)	19 (17.4%)	30 (27.5%)	24 (22.0%)	8 (7.3%)	4 (3.7%)
Do you feel ashamed of your skin when you are together with your sexual partner?	26 (23.9%)	16 (14.7%)	20 (18.3%)	31 (28.4%)	12 (11.0%)	4 (3.7%)
Do you experience rejection due to your skin condition?	42 (38.5%)	15 (13.8%)	30 (27.5%)	13 (11.9%)	6 (5.5%)	3 (2.8%)
Are you stressed before sexual intercourse because of your skin condition?	27 (24.8%)	22 (20.2%)	17 (15.6%)	25 (22.9%)	14 (12.8%)	4 (3.7%)

**Table 3 medicina-58-01443-t003:** Results regarding embarrassment and attractiveness level in studied patients.

	Not at All	Yes, a Little	Yes, Markedly	Yes, Very Much	Missing Data
Do you feel unattractive because of your skin disease?	6 (5.5%)	31(28.4%)	34 (31.2%)	36 (33.0%)	2 (1.8%)
Do you feel embarrassed when skin lesions occur on visible body areas?	5 (4.6%)	33 (30.3%)	30 (27.5%)	39 (35.8%)	2 (1.8%)
Do you feel embarrassed when skin lesions occur in the genital area?	15 (13.8%)	22 (20.2%)	38 (34.9%)	30 (27.5%)	4 (3.7%)
Has your sexual activity decreased because of the skin problem?	31 (28.4%)	41 (37.6%)	24 (22.0%)	9 (8.3%)	4 (3.7%)

**Table 4 medicina-58-01443-t004:** The influence of clinical and demographic parameters on sexual problems, quality of life and erectile dysfunction among studied patients with psoriasis (DLQI—Dermatology Life Quality Index, IIEF—International Index of Erectile Dysfunction, PASI—Psoriasis Area and Severity Index, PLN—Polish zloty).

	Sexual Problems	*p*	DLQI	*p*	IIEF	*p*
Age (years)	r = −0.03	0.77	r = −0.08	0.39	r = −0.33	0.01
Psoriasis duration (years)	r = 0.06	0.51	r = −0.02	0.87	r = 0.0	1
PASI (scores)	r = 0.2	0.04	r = 0.36	<0.001	r = −0.15	0.24
Gender						
-Female	20.6 ± 10.3	0.37	14.1 ± 8.7	0.26	-	-
-Male	18.8 ± 10.1		12.2 ± 8.7			
Education						
-Primary school	19.2 ± 12.6	0.71	11.4 ± 9.5	0.71	13.0 ± 6.0	<0.001
-Secondary school	21.4 ± 9.9		13.7 ± 8.0		18.3 ± 4.8	
-High school	18.8 ± 10.2		12.0 ± 7.9		22.1 ± 2.6	
-University	18.7 ± 10.0		14.0 ± 10.1		22.6 ± 4.9	
Domicile						
-Countryside	18.2 ± 10.9	0.71	13.0 ± 8.2	0.08	20.3 ± 4.6	0.26
-A city with up to 50,000 inhabitants	20.5 ± 7.9		9.7 ± 8.1		18.4 ± 6.3	
-A city with 50–150,000 inhabitants	20.3 ± 10.1		16.2 ± 6.6		18.4 ± 6.4	
-A city with over 150,000 inhabitants	20.7 ± 11.2		15.1 ± 9.9		22.1 ± 2.9	
Earnings						
-Earnings lower than PLN 2000	19.7 ± 9.9	0.72	12.2 ± 7.6	<0.05	15.9 ± 6.4	<0.01
-Earnings from PLN 2000 to 5000	20.0 ± 10.9		14.6 ± 9.4		21.4 ± 3.3	
-Earnings from PLN 5000 to 10,000	18.2 ± 7.4		6.7 ± 6.1		20.9 ± 5.3	
-Earnings above PLN 10,000	13.3 ± 6.7		9.3 ± 3.1		22.5 ± 3.5	
Marital status						
-Single	21.1 ± 10.4	<0.05	14.0 ± 8.9	0.27	18.6 ± 6.7	<0.001
-Married	18.2 ± 9.6		12.5 ± 8.7		21.3 ± 3.2	
-Divorced/Separated	25.8 ± 10.6		16.1 ± 9.2		19.1 ± 5.0	
-Widow/Widower	14.7 ± 10.7		8.3 ± 5.0		7.5 ± 2.1	
Cardiovascular disease						
-Yes	18.2 ± 9.8	0.43	11.0 ± 7.2	0.18	16.4 ± 5.9	<0.01
-No	20.0 ± 10.3		13.6 ± 9.1		20.9 ± 4.5	
Diabetes						
-Yes	18.8 ± 10.5	0.74	10.9 ± 7.8	0.29	18.3 ± 5.8	0.24
-No	19.7 ± 10.2		13.4 ± 8.8		20.4 ± 4.9	
Location of skin lesions						
-Face	22.4 ± 9.0	0.03	14.9 ± 8.5	0.1	19.6 ± 5.6	0.69
-Hands	21.2 ± 10.7	0.05	14.7 ± 9.0	0.02	20.1 ± 4.8	0.94
-Genital area	22.2 ± 10.3	0.01	15.8 ± 8.8	0.001	19.3 ± 5.8	0.4
-Scalp	19.6 ± 10.3	0.91	13.8 ± 8.9	0.09	20.6 ± 5.0	0.18
-Nails	20.5 ± 9.6	0.31	12.8 ± 8.8	0.83	20.7 ± 4.7	0.3
Type of psoriasis						
-Plaque-type psoriasis	19.9 ± 10.4	0.47	12.8 ± 9.0	0.75	20.3 ± 5.0	0.45
-Palmoplantar pustular psoriasis	17.1 ± 9.9	0.36	12.5 ± 8.7	0.86	20.3 ± 5.6	0.97
-Psoriatic arthritis	20.9 ± 7.5	0.54	12.3 ± 8.3	0.74	20.2 ± 3.7	0.9
-Nail psoriasis	18.9 ± 11.2	0.8	10.9 ± 8.6	0.33	22.8 ± 2.7	0.1
-Scalp psoriasis	19.0 ± 11.1	0.78	13.4 ± 9.1	0.78	23.2 ± 1.7	0.02
Family history of psoriasis						
-Yes	18.5 ± 10.3	0.36	12.7 ± 8.7	0.76	20.8 ± 5.0	0.27
-No	20.3 ± 10.1		13.2 ± 8.7		19.4 ± 5.1	
Patient’s assessment of psoriasis severity						
-Mild	11.5 ± 9.0	0.001	3.6 ± 3.4	<0.001	18.2 ± 6.2	0.63
-Moderate	16.2 ± 9.1		9.8 ± 7.9		20.4 ± 4.7	
-Severe	22.1 ± 9.9		15.6 ± 8.3		20.1 ± 5.1	

## Data Availability

Data are available from the corresponding author upon reasonable request.

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
