# Peer review of "Sexual Dysfunction in Women and Men with Psoriasis: A Cross-Sectional Questionnaire-Based Study"

_medicina, 2022, doi:10.3390/medicina58101443_

Round 1
Reviewer 1 Report
Dear Authors,
I found the manuscript wisely written and discussed. The way you presented the topic was great! Only few remarks: the table presents double sections (by mistake), and the part after the manuscript (from author contributions onwards) has not been completed.
Excellent job!
Author Response
We are grateful to the reviewer for the very supportive comments. We have corrected the table and amended the part after the manuscript.
Reviewer 2 Report
Thank you for submitting the manuscript. I have read your study with great attention and interest and I am very impressed by the subject. However, in my warning some are needed.1) The introduction does not mention pain and joint problems among the possible comorbidities of psoriasis, which instead are very important, also in terms of quality of life. In this regard, I suggest you read and add the following references:
doi: 10.1007/s11916-021-00952-5.
doi: 10.1177/0964528420920281.
doi: 10.2340/00015555-3906.
2) You don't mention the exact type of study you did. It should be specified better in the methods and in the title
3) The protocol number of the study and the Ethics Committee that approved it must be specified in the appropriate section.
4)You must complete all sections at the bottom of the manuscript in the style of the MDPI.
I hope these comments will be useful to you
Kind Regards
Author Response
1. In the introduction we have provided information about joint disease in psoriasis and have added 3 new references.
2. The type of the study was mentioned in the "Methods" section. The tile was changed accordingly.
3. The number of the Ethic Committee approval has been provided at the end of the manuscript.
4. All sections at the end of the manuscript has been completed as required by MDPI.
Reviewer 3 Report
This work studied sexual dysfunction in women and men with psoriasis. There are some issues in this manuscript that should be addressed as follows:
· Abstract:
- The meaning of the abbreviations should be clearly defined at their first mention (e.g. PASI, QoL).
- The overall conclusion should be mentioned clearly at the end of the abstract.
- There are many grammatical and typing errors in the abstract. Please, revise
- Key words: At least four key words should be mentioned.
· Introduction:
- The novel points in this study should be clarified in the “Introduction” section as there are many previous reports that discussed the same issue.
- The “Introduction” should be enriched by more recent references
· Materials and methods:
1. The ethical committee that approved this study and the ethical approval code should be mentioned.
2. How was the “11-item Questionnaire” validated?
3. The presence of other systemic diseases that may affect the sexual functions should be excluded.
· Results:
- A collective diagram summarizing the main findings of this study is recommended.
- Most of the results are descriptive in nature without delineating its statistical significance.
· Discussion: The discussion should provide more details to analyze of the results of the present study.
· Authors’ contribution should be mentioned.
· References: The number of the references is too small for a research article. At least 30 references may be acceptable.
· General comments:
1. The manuscript should be revised by English-naïve speaker to improve the quality of the language.
2. The manuscript should be checked regarding the grammatical errors and plagiarism.
3. The meaning of the abbreviations should be clearly defined at their first mention
Author Response
We are grateful to the reviewer for all constructive remarks. The following changes have been made:
- All abbreviations have been explained when used for the first time.
- In our opinion, conclusions have been mentioned at the end of the abstract. We want to ask the reviewer for being more specific if still something is missing.
- The Abstract has been revised in order to correct all spelling and grammar mistakes.
- The number of keywords was increased to four.
- We have provided a better justification for the study.
- We have provided new references in the Introduction.
- The ethics committee approval has been mentioned at the end of this manuscript.
- We have analyzed the influence of comorbidities on sexual life. However, as the number of particular comorbidities was small, we have only demonstrated the data for cardiovascular disorders. Indeed, we have observed, that patients with cardiovascular diseases had lower IIEF scoring. The other comorbidities did not influence significantly erectile dysfunction, but it may be biased by the low number of analyzed patients.
- Validation of the questionnaire was performed in the previous study (ref. 13). We tested the questionnaire for construct validity, internal consistency (Cronbach alfa=0.9), convergent validity, and ceiling and bottom effect.
- Regarding the results, we have analyzed statistically all studied parameters. Details are provided in tables.
- We have increased the number of references to 29.
- Although a graphical abstract is interesting, we are not convinced that it is really recommended by the journal guidelines.
- The manuscript has been carefully edited the manuscript for spelling and grammar mistakes.
Round 2
Reviewer 3 Report
The authors had appropriately addressed most of my comments
Author Response
The manuscript was again reviewed by a professional translator and some minor amendments have been provided to the previously revised manuscript.
In the table, we have provided data on diabetes.
We have provided a graphic abstract.
We will be grateful to the reviewer, if he/she could be more specific regarding the issues, which need to be solved.